# Artificially controlled nanoscale chemical reduction in VO$_2$ through electron beam illumination

Yang Zhang [1,5], Yupu Wang[2,5], Yongshun Wu[1], Xinyu Shu[1], Fan Zhang[1], Huining Peng[1], Shengchun Shen [1], Naoki Ogawa[3], Junyi Zhu[2] ✉ & Pu Yu [1,4] ✉

Chemical reduction in oxides plays a crucial role in engineering the material properties through structural transformation and electron filling. Controlling the reduction at nanoscale forms a promising pathway to harvest functionalities, which however is of great challenge for conventional methods (e.g., thermal treatment and chemical reaction). Here, we demonstrate a convenient pathway to achieve nanoscale chemical reduction for vanadium dioxide through the electron-beam illumination. The electron beam induces both surface oxygen desorption through radiolytic process and positively charged background through secondary electrons, which contribute cooperatively to facilitate the vacancy migration from the surface toward the sample bulk. Consequently, the VO$_2$ transforms into a reduced V$_2$O$_3$ phase, which is associated with a distinct insulator to metal transition at room temperature. Furthermore, this process shows an interesting facet-dependence with the pronounced transformation observed for the c-facet VO$_2$ as compared with the a-facet, which is attributed to the intrinsically different oxygen vacancy formation energy between these facets. Remarkably, we readily achieve a lateral resolution of tens nanometer for the controlled structural transformation with a commercial scanning electron microscope. This work provides a feasible strategy to manipulate the nanoscale chemical reduction in complex oxides for exploiting functionalities.

Chemical reduction is one of the most generic and essential reactions in solid state chemistry, which could drastically change the physical and chemical properties of functional materials[1,2]. In complex oxides, this reaction is usually dominated by the formation of oxygen vacancy[3], which plays a pivotal role in exploiting exotic crystalline structures with emergent phenomena. For instance, the incorporation of oxygen vacancies through chemical reduction forms an important pathway to manipulate the ground state of strongly correlated materials, as every oxygen vacancy would donate two electrons into the lattice. Importantly, when the oxygen vacancies form ordered lattice, it would lead to a distinct phase transformation from perovskite into brownmillerites[4,5] and even planar infinite layered structures[6–8] with distinct magnetic and electronic properties from the initial perovskite structures. Furthermore, the oxygen vacancy forms an indispensable ingredient for the manipulation of ionic conductivity[9] and catalysis[10] properties, as well as demonstrates approaches to design advanced piezoelectric[11] and magnetoelectric[12] couplings. Therefore, controlling the chemical reduction through the oxygen vacancy offers an exciting

[1]State Key Laboratory of Low Dimensional Quantum Physics and Department of Physics, Tsinghua University, Beijing 100084, China. [2]Department of Physics, The Chinese University of Hong Kong, Hong Kong SAR 999077, China. [3]RIKEN Center for Emergent Matter Science (CEMS), Wako 351–0198, Japan. [4]Frontier Science Center for Quantum Information, Beijing 100084, China. [5]These authors contributed equally: Yang Zhang, Yupu Wang. ✉e-mail: jyzhu@phy.cuhk.edu.hk; yupu@mail.tsinghua.edu.cn

and highly rewarded pathway to exploit functionalities. Conventionally, the thermal treatment within the reducing environment[4,13] or chemical agents[6–8] form the most accessible method to achieve this purpose as it can dramatically modify the thermodynamic phase diagram of the bulk compounds. However, this approach faces its fundamental limitation for functional miniaturization, as the whole sample would undergo the identical environment. Alternative approaches, using electric-field[14,15], ultraviolet light[16] or scanning tip[17,18], etc., have been proposed and demonstrated to manipulate the thermodynamic stability at the microscale, while nanoscale control is still of great challenge, as these methods are constrained by the corresponding intrinsic length scales.

To realize the nanoscale control, one should identify a pathway compatible with high spatial resolution and the capability of tuning the thermodynamic stability of materials. A method to simultaneously satisfy these requirements is the high energy electron beam (e-beam), in which the picometer wavelength of e-beam endows it with the ability to realize nanometer/sub-nanometer probe, and furthermore the lattice and electronic structure of materials could be significantly modified by incident electron beams, through various types (i.e., electron, photon and phonon) of excitations[19]. Among these interactions, the excited secondary and auger electrons contribute to the accumulation of positive charge at the sample surface, leading to a pronounced

internal electrostatic field at the sample surface. With the assistance of this electric field, the oxygen vacancy tends to migrate across the sample surface, leading to a dramatically enhanced oxygen vacancy evolution (Fig. 1a). Furthermore, through in-situ atomically-resolved imaging with the transmitted electron microscopy (TEM), the e-beam could also provide direct structural insight of phase transformation during multiple processes, like material decomposition[20], boundary migration[21] and topotactic transformation[22,23].

In this work, we exploit e-beam to induce chemical reduction in VO₂ with an accompanied insulator to metal transition (IMT) at the nanoscale lateral resolution. VO₂ possesses an IMT near room temperature from low-temperature monoclinic insulating phase to high-temperature rutile metallic phase[24,25], and more importantly vanadium oxides have a rich selection of oxidized phases with distinct properties. The IMT of VO₂ is strongly coupled with its stoichiometry, including the introduction of oxygen vacancy[26], hydrogen intercalation[27] and chemical doping[28]. Moreover, given the vast applications of VO₂ and its associated IMT for memory[29], electrochromic effect[30,31], metamaterial[32], optical modulator[33], etc., the manipulation of IMT at nanoscale forms a promising pathway to achieve improved performances. Our work highlights the capability of nanoscale control of chemical reduction assisted by the e-beam and offers possibilities to design functions through sketchable chemical reaction.

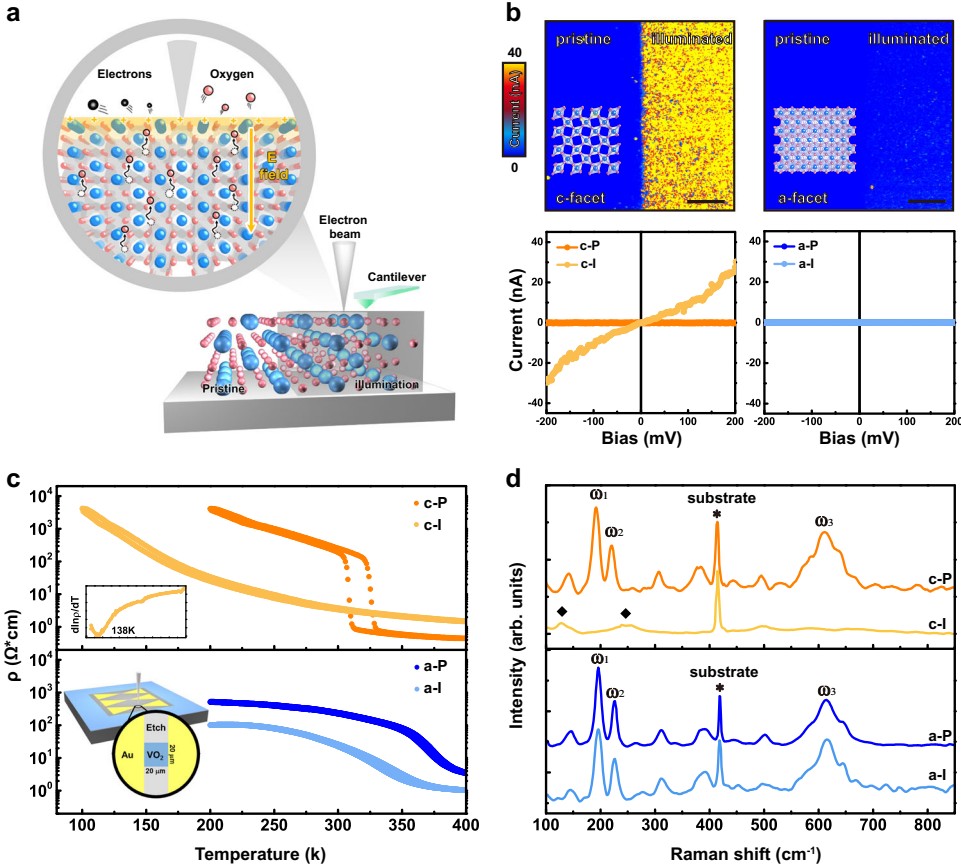

**Fig. 1 | Manipulation of the chemical reduction in VO₂ through e-beam illumination. a** Schematic illustration of the mechanism for the e-beam induced chemical reduction. The black circles represent the secondary electrons, which are created by the interaction with an incident electron beam and escapes from the surface area. The cantilever is used to characterize the local conductivity of the illuminated area. **b** Facet-dependence of the e-beam induced chemical reduction. Upper panel shows the current maps of the pristine and e-beam illuminated regions for c-facet and a-facet VO₂ samples. The superimposed images show corresponding atomic models of VO₂, as viewed along these two orientations. The scale bar is 2 μm. Lower panel demonstrates the characteristic local *I–V* curves measured at pristine (c-P and a-P) and illuminated (c-I and a-I) regions with a conducting AFM tip as the top electrode. **c** Temperature-dependent sheet resistances of both c- and a-facet VO₂ samples before and after e-beam illumination. The inset in the bottom panel shows a schematic device setup for the measurement. **d** Comparison of Raman spectra before and after e-beam illumination. To eliminate the background signal from TiO₂ substrates, VO₂ thin films grown on the Al₂O₃ (10−10) and (0001) substrates with the corresponding c- and a- facets were employed for the Raman measurements.

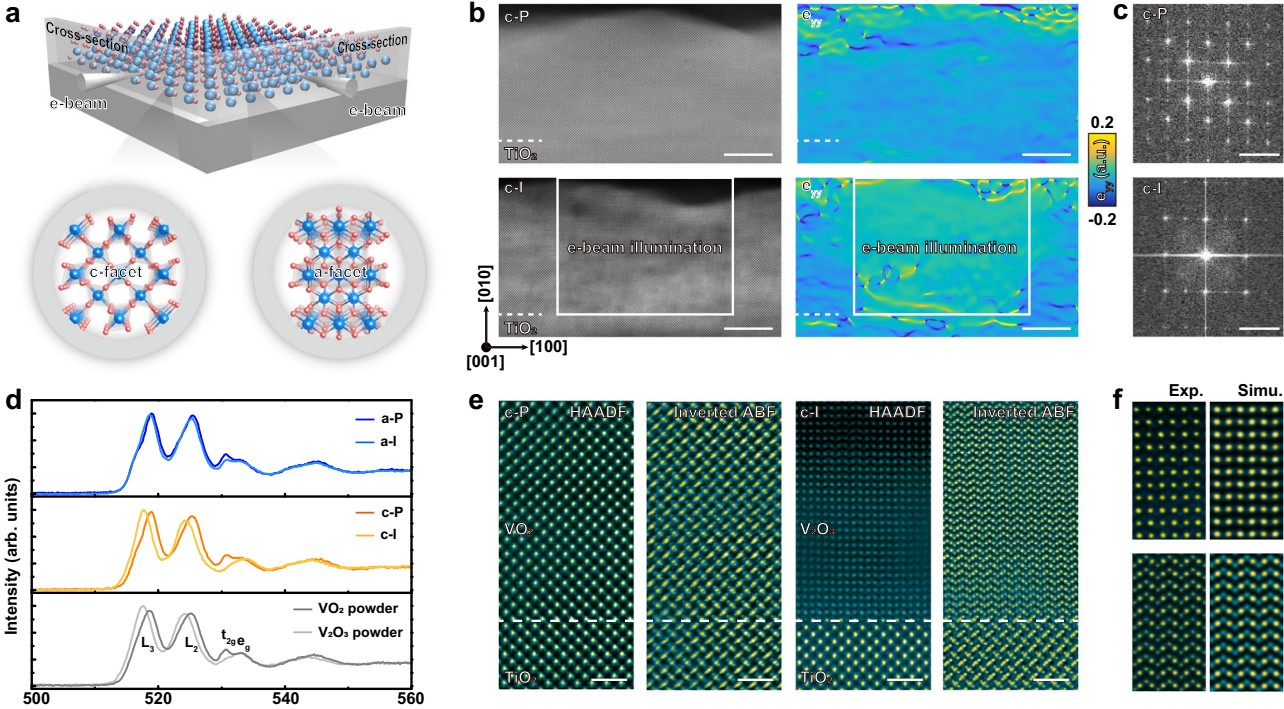

**Fig. 2 | Atomic insights for the e-beam induced chemical reduction in VO₂.**
**a** Schematic illustration of c- and a-facet VO₂ specimens prepared for STEM characterizations. **b** e-beam induced structural transformation in c-facet VO₂ sample. Left panel: Low-magnified HAADF images acquired at the same region before and after e-beam illumination. Right panel: Corresponding GPA analysis showing the emergence of a new crystalline structure after e-beam illumination. The scale bar is 20 nm. **c** FFT results collected from the pristine (c-P) and illuminated (c-I) regions. The scale bar is 5 nm⁻¹. **d** EELS measurements for a-facet and c-facet VO₂ samples before and after e-beam illumination. Reference spectra of $V_2O_3$ and $VO_2$ are collected from commercial powders. **e** High-resolution HAADF and inverted ABF images collected from pristine (c-P) and illuminated (c-I) regions of c-facet VO₂. The scale bar is 1 nm. **f** Left panel: Experimental HAADF and inverted ABF images of illuminated thin films. Right panel: Simulated ABF and HAADF images using rhombohedral $V_2O_3$ as input structure. The zone axis is [10–10] of rhombohedral $V_2O_3$.

## Results

We first demonstrate the feasibility of manipulating the chemical reduction in VO₂ through e-beam illumination. As the chemical reduction of oxides is strongly correlated with the surface facets, we prepared two different single-crystalline VO₂ thin films with non-polar and polar facets (termed as c-facet and a-facet), in which the VO₂ were deposited on $(001)_R$ and $(100)_R$ TiO₂ substrates respectively (methods and Supplementary Fig. S1). A conventional scanning electron microscopy (SEM) setup was employed to illuminate (scan) selected regions of these films (methods). Noting that the creation of oxygen vacancy will lead to a dramatically enhanced conductivity from the insulating VO₂ phase at room temperature, we carried out the measurements of local conductivity with the conducting atomic force microscopy (cAFM) on both samples (upper panel of Fig. 1b). The results reveal that a dramatically enhanced current emerges for the c-facet sample, while only subtle change is observed for the a-facet one. Such facet-dependent behavior can be further captured through local $I–V$ measurements, as exhibited at the lower panel of Fig. 1b. The conductivity is enhanced by more than two orders of magnitude for c-facet sample after illumination, while no measurable change is detected for the a-facet sample. We note that the surface morphology and roughness should not play a major role here because similar surface morphology is consistently observed in these two orientated thin films (Supplementary Fig. S2). Also, it is interesting to note that the Kelvin probe measurements reveals rather invisible change of the surface potential (Supplementary Fig. S3), which can then rule out the contribution of charge inject through e-beam illumination.

The facet-dependent IMT feature observed in cAFM also reflects in the temperature-dependent electrical resistive measurements

(Fig. 1c). After e-beam illumination, the characteristic IMT (at ~325 K) for pristine c-facet VO₂ completely disappears and the sample resembles a semiconducting behavior with an anomalous transition at ~138 K (upper panel of Fig. 1c). However, the illuminated a-facet sample still possesses a IMT feature close to its pristine transition temperature, with a slight decrease of resistivity and a wider thermal hysteresis (lower panel of Fig. 1c and Supplementary Fig. S4). To further reveal the structural change of these two facets through e-beam illumination, we carried out Raman spectroscopy. Figure 1d reveals the characteristic Raman peaks for pristine VO₂ at 194 cm⁻¹ ($\omega_1$), 223 cm⁻¹ ($\omega_2$) and 612 cm⁻¹ ($\omega_3$), which are attributed to the V–V dimer interaction and different V-O bond lengths in monoclinic insulating $VO_2$[34]. For illuminated c-facet VO₂ sample, these characteristic peaks are dramatically suppressed, and two new peaks (marked by diamonds) emerge, which provides direct evidence for e-beam induced structural transformation. Nevertheless, for a-facet VO₂, the characteristic peaks remain intact even through e-beam illumination, indicating that the a-facet sample still maintains its monoclinic structure.

To provide further structural insights for the phase transformation, we employed in-situ scanning transmitted electron microscopy (STEM) to trace the structural evolution through e-beam illumination. To clarify the facet dependent behavior, we prepared two cross-sectional TEM samples with the zone axis normal to c- and a-facet, respectively (Fig. 2a). We used scanning e-beam in STEM mode to illuminate these two samples, and then the atomically-resolved STEM images were simultaneously captured. As shown in Fig. 2b, a new phase can be readily recognized at the illuminated area of c-facet VO₂, showing an elongated out-of-plane lattice constant according to the geometric phase analysis (GPA). Such structural transformation is

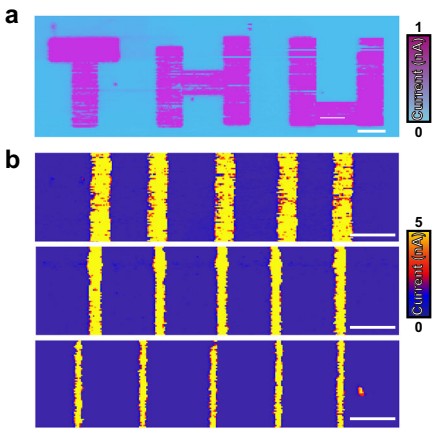

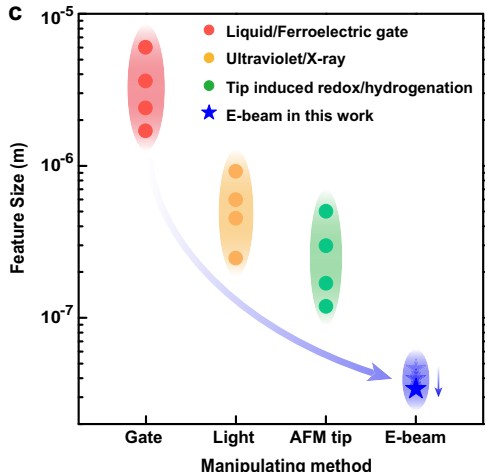

**Fig. 3 | Nanoscale phase transformation through e-beam illumination. a** Current map of a designed pattern with the "THU" template. The scale bar is 1 μm. **b** Current maps of a series of stripes with different widths. The scale bar is 500 nm.

**c** Comparison of characteristic feature sizes induced from insulating $VO_2$ samples through different methods. The reference data are adapted from literatures[14,55–65].

further supported by the FFT results and selected area diffraction patterns (Fig. 2c and Supplementary Fig. S5). As for a-facet $VO_2$, the structural feature of monoclinic $VO_2$ remains almost unchanged under the same dose of e-beam illumination (Supplementary Figs. S6), consistent with the results depicted in Fig. 1.

Electron energy loss spectroscopy (EELS) was used to determine the valence state of vanadium ions within this new phase. Figure 2d summarizes EELS signals at vanadium $L$-edges and oxygen $K$-edges collected from the pristine and illuminated regions within c- and a-facet samples, while the evolution with the illuminating dose is shown in Supplementary Fig. S7. For signals collected from illuminated c-facet sample, a noticeable "red shift" of vanadium $L_3$-edge is observed. By comparing with the EELS signals collected from commercial $VO_2$ and $V_2O_3$ powders (Supplementary Fig. S8), it suggests a reducing valence state from $V^{4+}$ to $V^{3+}$, occurs in c-facet sample after e-beam illumination. This assignment is further supported by the dramatically suppressed peak intensity at $t_{2g}$ peak of oxygen $K$-edge, which is contributed by the transition from O-1$s$ to the hybridization between O-2$p$ and V-3$d$ ($d_{xz}$, $d_{yz}$) state[35]. On the other hand, the illuminated a-facet sample shows a much smaller change of peak position and intensity at $L$-edges and oxygen $K$-edges respectively, which is consistent with the fact that the a-facet sample remains its pristine monoclinic phase with small amount (about 3%) of oxygen vacancy (Fig. 1c, d).

To elaborate the crystalline structure of the new phase transformed from the c-facet $VO_2$, we collected high-resolution STEM image as shown in Fig. 2e. Combined with the high angle annular dark field (STEM-HAADF) and annular bright field (STEM-ABF) image, the atomic positions of vanadium and oxygen ions are determined, which together with the simulation result (Fig. 2f), providing compelling evidence that the illuminated area transforms from monoclinic $VO_2$ to rhombohedral $V_2O_3$. Remarkably, the $V_2O_3$ is found to be fully epitaxial with the substrate (Supplementary Fig. S9–S10). This assignment can nicely explain the observed $R−T$ curves and Raman spectra shown in Fig. 1, as similar $R−T$ curves were reported in pristine $V_2O_3$ thin films[36,37], and the characteristic Raman peaks in the illuminated c-facet sample could be attributed to the features of rhombohedral $V_2O_3$ (Supplementary Fig. S11)[38].

We then address the mechanism of the e-beam-induced phase transformation in c-facet $VO_2$ sample. We summary various possible interactions between the e-beam and materials in Supplementary Fig. S12, which can be classified into knock-on, radiolysis, thermal heating as well as the electrostatic field[19]. Firstly, the knock-on effect,

which is due to collision between the high energy electrons and heavy ions, was widely reported during the TEM measurements as the e-beam penetrating through the thin film; while this effect is less notable for SEM measurements, in which the beam cannot penetrate through the specimen. The fact that e-beam induced phase transformation was observed through both SEM and TEM methods, can then rule out the contribution of knock-on effect. Secondly, the thermal heating effect, which is caused by the phonon excitation, can also be excluded based on the in-situ XRD measurements (Supplementary Fig. S13), which reveals that the $VO_2$ phase remains robust with the temperature up to 170 °C in vacuum. While considering the rather small dose of electrons used, we expect the sample temperature is much less than 170 °C, and then the influence of thermal heating can be excluded. The radiolysis and internal electrostatic field are hard to be decoupled, as the former is able to induce oxygen desorption from the surface[39], which creates a gradient chemical potential for oxygen ions between the bulk and the surface[40,41], while the latter would naturally facilitate the migration of oxygen vacancy from surface toward the sample bulk due to positively charged surficial area[42]. Eventually, we speculate that these two mechanisms would contribute cooperatively to facilitate the observed phase transformation from $VO_2$ into the corresponding $V_2O_3$ (see Supplementary Materials for detailed discussion).

After the demonstration of the working principle of e-beam induced chemical reduction, we then focus on investigating its high spatial resolution. For this propose, a "THU" pattern was adapted as a lithographic template for the e-beam illumination through a conventional SEM setup on a c-facet $VO_2$ sample. The corresponding conducting pattern (Fig. 3a) follows nicely with the designed pattern, which showcases the feasibility of imprinting artificially designed high precise pattern through e-beam. We further investigated the spatial resolution of the e-beam induced transformation by focusing the electron beam with a SEM setup. As shown in Fig. 3b, the characteristic size can be controlled nicely, whose lateral size is estimated to be from ~250 nm to ~50 nm (Supplementary Fig. S14). We note that this feature size is much smaller than the other methods, as summarized in Fig. 3c. Therefore, we demonstrate the e-beam illumination as a promising pathway to realize nanoscale control of chemical reduction.

To provide further understanding of the facet-dependent e-beam induced phase transformation, we carried out density functional theory (DFT) calculations. We considered the evolution of oxygen vacancies into three processes: the formation of oxygen vacancies on the surface, the insertion from the surface to the bulk, and then the

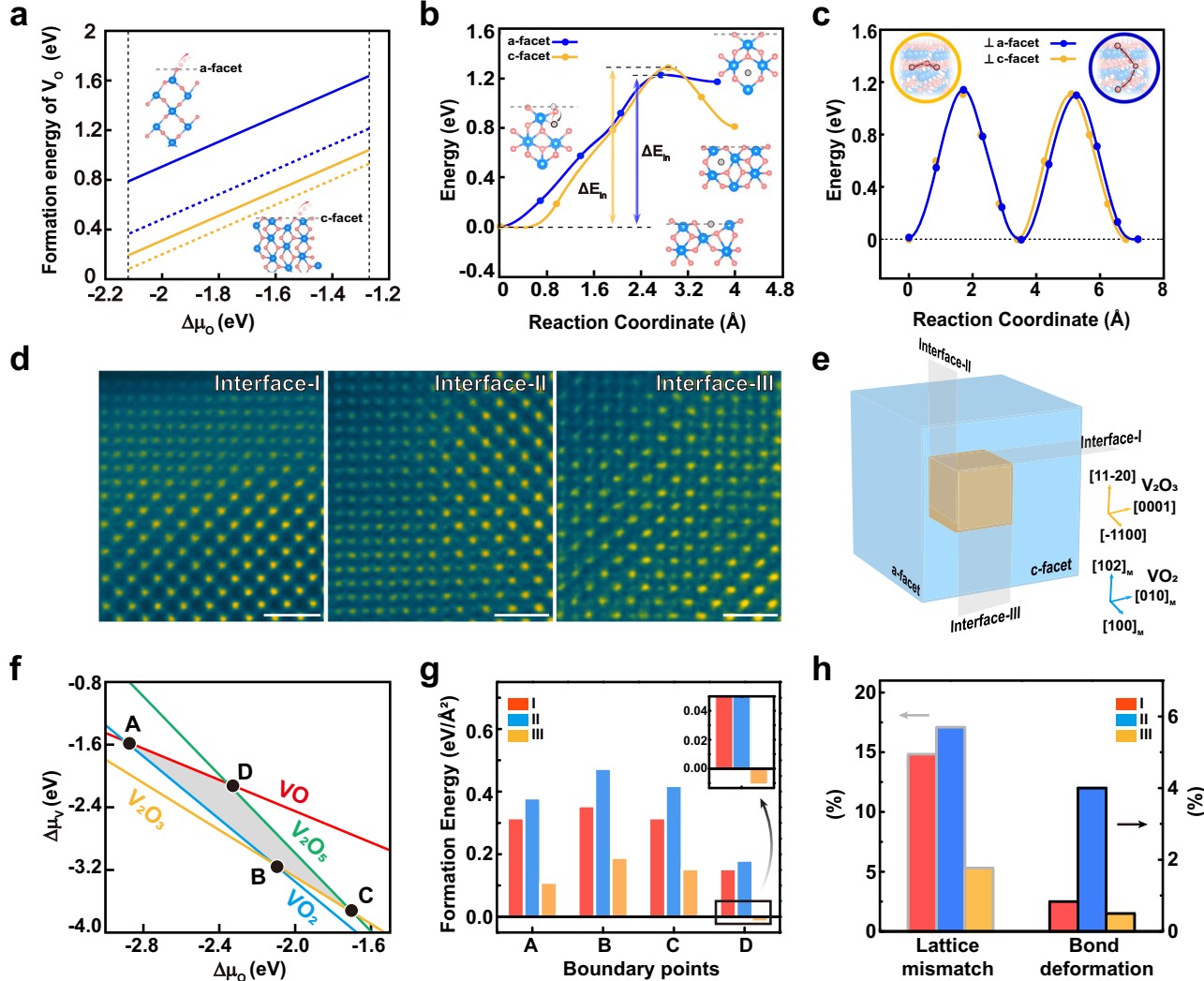

**Fig. 4 | Mechanism of facet-dependent chemical reduction and phase transformation in VO₂. a** Theoretically calculated oxygen vacancy formation energy for a- (blue) and c-facet (yellow) VO₂. The solid and dashed lines represent results calculated through GGA and HSE functional methods, respectively. The insets show schematic illustrations of the formation of oxygen vacancy at these two surfaces. **b** Energy profiles for oxygen vacancy diffusion from the surface into the bulk. ΔE$_{in}$ is the barrier of vacancy diffusion from surface to sub-surface. The inset shows the initial and final states during vacancy intercalation, where the gray circle represents the oxygen vacancy. **c** Energy profiles for oxygen vacancy diffusion along two different directions. The insets show the corresponding diffusion paths. **d** HAADF images of three characteristic interfaces. The scale bar is 1 nm. **e** Schematic illustration of three VO₂/V₂O₃ interfaces as observed by STEM after e-beam illumination on c-facet sample. **f** Theoretically calculated phase diagram for vanadium oxides, while the intersections between VO₂ and other oxides are labeled as A, B, C, D, respectively. **g** Calculated formation energies for three different VO₂/V₂O₃ interfaces at the boundary points of phase diagram. **h** Associated lattice mismatch and V-O bond deformation at three different VO₂/V₂O₃ interfaces.

diffusion within bulk. We first computed the formation energy of oxygen vacancies on both facets with the most stable reconstructions (Supplementary Fig. S15), as shown in Fig. 4a. The results obtained from both GGA and static HSE functional methods, reveal conclusively that the c-facet has a much lower formation energy than a-facet, suggesting that it is easier to create oxygen vacancy at the c-facet, likely through the e-beam induced oxygen desorption. We then calculated the migration barriers of the oxygen vacancies from surfaces toward the bulk for both facets, and got comparable energies, as shown in Fig. 4b. Finally, we computed the energy barriers within the bulk along normal to a- and c-facet (Fig. 4c and Supplementary Fig. S16), which reveal a comparable barrier along these two directions. These calculated results then clearly suggest that the formation of the oxygen vacancies at the sample surface is the rate limited factor for the phase transformation, and along this vein the c-facet sample is inclined to be reduced. Furthermore, we also expect that the e-beam induced positively charged background through the creation of secondary

electrons at the sample surface leads to an effective internal electrostatic field to facilitate the formation of positively charged oxygen vacancies at the sample surface as well, which then migrates gradually into the bulk to trigger the phase transformation.

The phase transformation can also be understood collectively by a comparison of VO₂/V₂O₃ interface stability along different directions, because the interfaces serve as the wavefront for the oxygen ion migration during the phase transformation. According to the STEM results (Fig. 4d), we identified three possible sharp interfaces (denoted as Interface-I, -II and -III) during phase transformation (Supplementary Fig. S17). These three interfaces represent three different terminations of VO₂ when it is montaged with V₂O₃. Specifically, the interface-II and -III represents the a- and c-facet respectively, while interface-I is perpendicular to [102]$_M$ direction (Fig. 4e). Within the parameter space allowing the coexistence of both VO₂ and V₂O₃ (Fig. 4f and Supplementary Materials for details), we calculated the formation energy for all three interfaces and found out that the interface-III is the most

stable one (Fig. 4g). This result strongly suggests that the direction of phase transformation prefers perpendicular to the interface-III (c-facet), consistent with our experimental results. This result can be understood through two correlated factors, i.e., lattice-mismatch and interfacial reconstruction (Fig. 4h). The calculations reveal that in interface-I and -II, the lattice-mismatches are much stronger than the interface-III (Supplementary Table S1). Moreover, the interface-I and -II also show larger lattice reconstructions (Supplementary Fig. S18 and Supplementary Tables S2–S3). Noting that interfaces-I and -II are along the polar directions, while the interface-III is along the nonpolar direction, this facet dependence is rationalized by the fact that the electron counting model[43–45] can be easily satisfied near the nonpolar surface and interface[46].

In summary, we demonstrate a tunable nanoscale chemical reduction at $VO_2$ via e-beam illumination, which is accompanied by a structural transformation into $V_2O_3$. We envision that this work not only opens a promising pathway to exploit functionalities related to chemical reduction at complex oxides, but also provides a strategy to in-situ investigate the oxygen ionic migration driven by electron beam illumination.

## Methods

### Growth of film
$VO_2$ thin films were grown with a customized pulsed laser deposition system at a growth temperature of 420 °C and an oxygen pressure of 0.8 Pa. The energy density of the laser ($\lambda = 248$ nm) was fixed at 1.8 J/cm$^2$ at the surface of ceramic $VO_2$ target, with a repetition rate of 3 Hz. After the growth, the samples were cooled down to room temperature with a cooling rate of 10 °C per minute at oxygen pressure of 2.0 Pa. The sample thickness was controlled by the growth time at a calibrated growth rate and then further confirmed with X-ray reflectometry measurements. The x-ray diffraction and reflectometry measurements were carried out using a high-resolution four-circle X-ray diffractometer (Smartlab, Rigaku) with monochromatic Cu Kα1 radiation ($\lambda = 1.5406$ Å). The thickness of thin films used for e-beam illumination, AFM characterization, electrical and Raman measurements is about ~18 nm, while slightly thicker (~50 nm) samples were employed for TEM experiments to minimize the influence of surface damage during sample preparation.

### Electron beam illumination
The e-beam illumination characterization was carried out with a scanning electron microscope (Zeiss-Merlin) under accelerate voltage of 30 kV and probe current of 30 nA. The illuminated area used for the conducting atomic force microscope (c-AFM) characterization was $8 \times 4$ μm$^2$ with a total electron dose of ~10$^5$ e/Å$^2$. While the samples used for electrical and Raman measurements were $20 \times 20$ μm$^2$ in dimension, with a total electron dose of ~$5 \times 10^5$ e/Å$^2$. The electron-beam illumination in transmission electron microscope (TEM) was carried out under the accelerate voltage 300 kV with 200 pA probe current. The illuminated region was $50 \times 50$ nm$^2$, which leads to an electron injection dose of ~10$^7$ e/Å$^2$.

### Scanning probe microscopy
The scanning probe microscopy (SPM) measurements, including morphology, local current and surface potential characterizations, were performed by a commercial scanning probe microscope (Cypher ES, Oxford Instruments), which was armed with Pt-coated conductive cantilevers (HQ: NSC18/Pt, MikroMasch) with a spring constant of ~2.8 N/m and a free resonance frequency of ~75 kHz. The conducting atomic force microscope (c-AFM) measurements were conducted to probe the local conductivity of the samples, in which the orca tip holder was selected. All conducting atomic force microscope mapping measurement were conducted with reading voltage of 0.8 V.

### Electrical measurement
Electrical transports were performed with the Physical Property Measurement System (PPMS, Quantum Design) by applying a direct current of 0.5 μA. The bridge $VO_2$ devices with a dimension of $150 \times 20$ m$^2$ were patterned through photo lithography process, followed by Ar beam etching (at calibrated etching rate) to remove the uncovered part. Ti/Au electrodes were subsequently coated on both ends of the $VO_2$ bridges with an uncovered region of $20 \times 20$ μm$^2$ at the middle for the subsequent electron beam illumination.

### Raman spectroscopy
Raman spectra were measured at ambient condition using a Horiba Jobin Yvon LabRAM HR Evolution spectrometer with the $\lambda = 514$ nm excitation source from an Ar laser and an 1800 gr/mm grating.

### Transmission electron microscopy
The cross-sectional TEM specimens were thinned down to less than 30 μm first by using mechanical polishing, and then argon ion milling was carried out using PIPSTM (Model 691, Gatan Inc.) with the accelerating voltage of 3.5 kV until a hole was drilled. Low voltage milling was performed with accelerating voltage of 0.3 kV to minimize the damage and remove the amorphous surface layer. Plasma cleaning was employed for all TEM samples to remove the free carbon adsorbed on the surface. The STEM-HAADF and STEM-ABF images were acquired on an FEI Titan Cubed Themis 60–300 (operated at 300 kV), capable of recording high-resolution STEM images with the spatial resolution of ~0.059 nm. The microscope was equipped with a high brightness electron gun (X-FEG with monochromator), a spherical aberration corrector, and a post-column imaging energy filter (Gatan Quantum 965 Spectrometer). The energy resolution was smaller than 0.3 eV. The collection angle of the HAADF and ABF detector was 48–200 mrad and 8–57 mrad, respectively. The image simulation was carried out by Dr. Probe, with 18 slices along [10–10] crystal axis of $V_2O_3$ were divided for each unit cell. The collection angle of HAADF and ABF detector was the same as the experimental setup. The Dual-EELS mode was selected to collect core-loss signals of both vanadium and oxygen ions. The entrance aperture was 5 mm and the step size was 0.1 eV. Data processing, including calibration of the zero-loss position, pre-edge background subtraction, removal of multiple scattering, and normalization, was carried out to extract EELS signals. All EELS data were processed using the Gatan Microscopy Suite 3.0 software package. Commercial $VO_2$ and $V_2O_3$ powders (Sigma Aldrich 99.5% pure) were used to collect the reference EELS spectra.

### Density functional theory calculation
Spin-polarized DFT calculations[47,48] implemented by Vienna Ab-initio Simulation Package (VASP)[49–51] were carried out during the study. Within the calculations, we used Perdew-BurkeErnzerhof (PBE)[52] formulation of the generalized-gradient approximation (GGA) for exchange-correlation function using projector augmented-wave method (PAW)[49–51] for the surface formation energy, the climbing image nudged elastic band (CI-NEB) method for migration barrier calculation and interface formation energy calculations. The volumes of simulation cells are fixed, and all atoms are allowed to relax. The minimization was adopted by the conjugate gradient algorithm. During the structural optimization, both the cell volume and atom positions are allowed to relax and move during the self-consistent calculation. While during the calculations of the surface formation energy, NEB, and interface formation energy calculation, the cell volume is fixed, while all atoms are allowed to relax. In the calculations, the electronic minimization adopted the blocked Davidson iteration scheme, and the ionic relaxation was used in the conjugate gradient algorithm. We also employed static hybrid Heyd-Scuseria-Ernzerhof (HSE) function with the structure using GGA method[53]. The empirical range parameter $\omega$ and the mixing ratio of exact exchange $\alpha$ in a

Hartree-Fock were set to be $0.2\,\text{Å}^{-1}$ and 0.1, respectively. For GGA and HSE calculations, the energy cutoff of plane waves was set as 450 eV and Feynman-Hellman forces were converged to less than 0.01 eV/Å. For the calculations of oxygen vacancy formation energy, we adopted Gamma-centered method and the k-point meshes of a- and c- facet samples were set to be $4 \times 4 \times 1$ and $5 \times 3 \times 1$, respectively. To account for the diffusion within the bulk, we adopted a 96-atom $(2 \times 2 \times 2)$ supercell and k-point mesh was set as $3 \times 3 \times 1$. To identify the diffusion paths, we used the climbing image nudged elastic band (CI-NEB)[54] method to map out the transition states, and the spring constant employed was $5\,\text{eV/Å}^2$.

## Reporting summary

Further information on research design is available in the Nature Portfolio Reporting Summary linked to this article.

## Data availability

All data supporting the results of this study are available in the manuscript or the supplementary information. Additional data are available from the corresponding author upon reasonable request.

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

## Acknowledgements

This study was financially supported by the National Key R&D Program of China (grant No. 2021YFE0107900); the Basic Science Center Program of NSFC (grant No. 52388201); the Beijing Nature Science Foundation (grant No. Z200007); the National Key R&D Program of China (grant No. 2021YFA1400300); the National Natural Science Foundation of China (grant Nos. 52025024 and 12104250) and the China Postdoctoral Science Foundation (grant Nos. 2022M721769 and 2022M711870). This work was also supported by the Research Grants Council of Hong Kong for General Research Fund (grant Nos. 14307018, 14307219 and 14319416) and research fund from SIAT-CUHK Joint Laboratory of Photovoltaic Solar Energy. Y.Z. acknowledges support from the Shuimu Tsinghua Scholar Program of Tsinghua University. X. S acknowledges support from International Postdoctoral Exchange Fellowship Program of China (Talent-Introduction Program).

## Author contributions

P.Y. and Y.Z. conceived the study. Y.W. prepared the $VO_2$ thin films. Y.Z. performed the e-beam illumination, Raman and TEM measurements. X.S. and S.S. performed the transport measurements. Y.Z. performed the cAFM measurements with the assistant from F.Z. and H.P., Y.W. carried out the DFT calculations supervised by J.Z., N.O. provided insights about the phase transformation. Y. Z., Y.W., J.Z. and P.Y. wrote the paper, while all authors discussed the results and commented on the manuscript.

## Competing interests

The authors declare no competing of interests.
