## [Peer Review File · Nature Communications]

Artificially controlled nanoscale chemical reduction in VO₂ through electron beam illuminationEditorial Note: Parts of this Peer Review File have been redacted as indicated to remove third-party material where no permission to publish could be obtained.

REVIEWER COMMENTS

Reviewer #1 (Remarks to the Author):

Dear Editor, the manuscript by Zhang et al describes the methodology and characterisation of e-beam illuminated VO₂ as a patterning technique. The authors make a compelling case to create a pattern of reduced V₂O₃ by using an electron beam in a matrix of VO₂, which appears also dependent on the orientation of the thin films. Experimentally, the work is very solid and I have no objections. It could be interesting for a broad audience and find more broad applications. To my knowledge, this specific effect and methodology has not been used in this manner before, although ebeam damage effects have been studied in more detail.

In the analysis and modelling, the authors consider interfacial activity as the main reason for a difference between the two investigated orientations. The authors, should however, also consider and show that morphological differences (grain boundaries) could play a role here. I recommend that the authors show the morphological differences between the two types of samples. If this can be included as well as a discussion, I heartily recommend publication in Nat. comm.

Reviewer #2 (Remarks to the Author):

The authors report a reduction of VO₂ to V₂O₃ through intense electron beam irradiation. first of all, the text is full of grammar and wording mistakes, so it has to be thoroughly read through and edited by a native English speaker. (for exemple “e-beam illustration” used multiple times on page 5 should be “e-beam illumination” and “ultraviloate” instead of “ultraviolet” on figure 3c)

Could the authors please comment on the O-stoichiometry of their pristine film as the phenomenon they report can strongly depend on the crystalline quality of their original film ? Electron dose rates are usually reported in electron/angstrom².s, could the authors convert it to this unit?

It would be useful that the authors compare the eels spectra acquired on commercial (fresh, prepared in dry conditions) powders of VO₂ and V₂O₃ to compare with their films as the sample preparation with ion beam irradiation might already damage the film and create oxygen vacancies. As the state of the material in its pristine is crucial to the claim of authors (displacement or creation of oxygen vacancies?)

As claimed by the authors, the films used for -beam illumination, AFM characterization, electrical and Raman measurements should be strained (18 nm case) but the ones used for TEM should be mostly relaxed (50 nm case). This structural change should have influence on the EELS if we are looking at very fine details across the thickness of the film, could the author provide depth dependent eels measurements across their pristine and irradiated films to verify the evolution of the V L_{2,3} and the O K edge across the thickness of the film ? X-ray reciprocal space maps could shine more light on the exact strain state of the films for the thinner cases.

In figure 2d, could the author verify the absolute energy scale of their spectra, it seems to be shifted, on the other hand on this figure as well as figure S6 in the supplementary material it is very difficult to see settle differences between the EELS spectra due to the large gap between the curves, for the purpose of fair comparison, it would be better to overlay the spectra and the authors should add the spectra of the pristine material in both orientations has there will be subtle differences that are important for the interpretation of the oxygen vacancies claim, as recently reported by Lee et al. (Science 2018)

The authors said that knock-on damage can be excluded to be the source of the observation, but as the microscope used for the STEM measurement can be used from 60 to 300kV, it would be interesting to see if the authors can reproduce the same effect observed at 300kV (in ABF/HAAADF images and in EELS) at a lower voltage (60kV) where knock-on damage will be strongly reduced. In general the discussion of the e-beam induced phase transformation should be developed in more details and documented by a carefully researched references in the literature as electron beam damage is a very complex topic. For example, I am not convinced by the statement of the authors stating that they exclude radiolysis as it would amorphized the specimen.

Concerning the facet dependence of the phase transition, as the sample prepared for TEM is a cross section of the specimen, it would be adequate to mention the crystallographic orientation next to each STEM image. Could the authors provide an image of the cross section of the film after e-beam irradiation of the 18nm film taken under one of the irradiated area from c-facet features presented in figure 3?

The sentence "Furthermore, we speculate that the e-beam induced positively charged background at the sample surface leads to an effective internal field as the thermodynamical driving force, boosting the diffusion of the negatively charged oxygen ions from the bulk toward sample surface." Is quite confusing to me because this could be understood as oxygen vacancies starting to form in the bulk rather than at the surface which is not observed in the stem images.

When looking at Figure S1, only the top left panel shows Laue fringes around the film peak which attests to sharp interfaces and is a good proxy for high film quality. It is disappointing to see that the samples on TiO₂ (which have a better epitaxial match than those on Al₂O₃) don't show such finite thickness fringes, could the authors comment on this.

Figure S5, also the TiO₂ substrate gets damaged? What is the reaction there? Some areas of the c-facet film are dark on the second image and white on the third, could the authors comment on this ? The surface of the film seems strongly damaged on the second image but recovers in the third image, could the authors comment of this ? I also see an increase of the amount of white spots between the first and second image of the a-facet but they become less visible on the third one, could the authors comment on this. This image has furthermore extremely low quality (similar to figures S4, S3, S6 and S7)

It would be also interesting to know how the authors avoided contamination with such long electron beam irradiations as hydrocarbons contamination is a well known side effect of electron beam illumination.

Reviewer #3 (Remarks to the Author):

In this manuscript, a method based on e-beam is exploited to induce chemical reduction in VO₂, which is accompanied by a structure transition into V₂O₃.

I think that the evidence to show the transition into V₂O₃ is very indirect, which is not very convincing.

1. In Fig. 1d, the Raman spectrum for C-R is so broad and it looks almost like an amorphous material. Only the peak from the substrate remains sharp, and it is not really convincing to claim the existence of the crystal structure.

2. It is also well-known that Vanadium oxide thin films could undergo various surface reconstruction. The EELS spectra of c-R exhibit a significant shift compared to the reference spectra of V₂O₃, which requires further investigations using other techniques like HAXPES.

3. For the DFT results, the criterion is set to be 0.02 eV/Å, which seems to be higher than typical recommended value of 0.01. Moreover, details about how the structural optimization

are missing. For example, are all atoms allowed to move? Is volume allowed to change? What optimization schemes in VASP are used?

4. V₂O₃ is known to have magnetic phase transition. It will be necessary to have some magnetic property analysis if the authors want to claim the existence of V₂O₃.

In summary, the present experimental data are not convincing enough to support the claim of the paper.

We thank all reviewers for their constructive suggestions and comments, which are very valuable in improving the quality of our manuscript.

Reviewer #1

Dear Editor, the manuscript by Zhang et al describes the methodology and characterization of e-beam illuminated VO₂ as a patterning technique. The authors make a compelling case to create a pattern of reduced V₂O₃ by using an electron beam in a matrix of VO₂, which appears also dependent on the orientation of the thin films. Experimentally, the work is very solid and I have no objections. It could be interesting for a broad audience and find more broad applications. To my knowledge, this specific effect and methodology has not been used in this manner before, although e-beam damage effects have been studied in more detail.

In the analysis and modelling, the authors consider interfacial activity as the main reason for a difference between the two investigated orientations. The authors, should however, also consider and show that morphological differences (grain boundaries) could play a role here. I recommend that the authors show the morphological differences between the two types of samples. If this can be included as well as a discussion, I heartily recommend publication in Nat. comm.

Response : We would like to express our gratitude to the reviewer for the positive recognition of our work and constructive comments. In this response, we performed atomic force microscopy (AFM) measurements on both two orientated VO₂ thin films. As shown in **Figure R1**, the AFM results indicate no measurable difference in terms of morphology and roughness. Furthermore, we note that the facet-dependent phase transformation behavior was also reproduced with the TEM, where the specimens were prepared by mechanical polishing and argon ion milling, resulting similar surface morphology between the samples. Therefore, we believe that the sample morphology does not play a major role for the observed facet-dependent phase transformation. In the revised manuscript, we added a sentence (**page 4, line 93**) “*We note that the surface morphology and roughness should not play a major role here because similar surface morphology is consistently observed in these two orientated thin films (Supplementary Fig. S2)*”. **Figure R1** was added as in the SI as **Figure S2**.

Figure R1. Atomic force microscope images of pristine a- and c-facet VO₂ thin films. The scale bar is 1 μm.

Reviewer #2

Q1. The authors report a reduction of VO₂ to V₂O₃ through intense electron beam irradiation. First of all, the text is full of grammar and wording mistakes, so it has to be thoroughly read through and edited by a native English speaker. (for example “e-beam illustration” used multiple times on page 5 should be “e-beam illumination” and “ultraviloate” instead of “ultraviolet” on figure 3c)

Response: We express our gratitude to the reviewer for carefully reviewing our work, and apologize for the poor writing. In this revised manuscript, we have incorporated the mentioned corrections and edited the manuscript.

Q2. Could the authors please comment on the O-stoichiometry of their pristine film as the phenomenon they report can strongly depend on the crystalline quality of their original film?

Response : To determine the oxygen stoichiometry, we carried out X-ray photoelectron spectroscopy (XPS) measurements on pristine VO₂ thin films with both facets, and then compared the obtained spectra with the reference data of V⁴⁺ ion. As shown in **Figure R2**, no significant discrepancy is observed between these spectra, and the obtained characteristic peaks on our samples resemble nicely the features of V⁴⁺ ion (Ferran, U.-B. *Appl. Sur. Sci.* **403** 717-727 (2017)), suggesting that the excellent chemical stoichiometry of our VO₂ thin films.

[Redacted]

Figure R2. XPS measurements of pristine c- and a-facet VO₂ thin films. The reference spectra of V⁴⁺ was adapted from reference (Ferran, U.-B. *Appl. Sur. Sci.* **403** 717-727 (2017)).

Q3. Electron dose rates are usually reported in electron/angstrom².s, could the authors convert it to this unit?

Response: We thank the reviewer for this suggestion. In this revised version, the unit of electron dose has been converted into e/Å² in both the main text and SI.

Q4. It would be useful that the authors compare the eels spectra acquired on commercial (fresh, prepared in dry conditions) powders of VO₂ and V₂O₃ to compare with their films as the sample preparation with ion beam irradiation might already damage the film and create oxygen vacancies. As the state of the material in its pristine is crucial to the claim of authors (displacement or creation of oxygen vacancies?)

Response: We appreciate the reviewer’s insightful suggestions. We prepared two powder samples (Sigma Aldrich 99.5% pure) and collected the EELS spectra, as shown in **Figure R3**. Clearly, the spectra taken from VO₂ and V₂O₃ powders resemble closely the spectra taken from pristine c-facet sample and the sample after e-beam illumination, respectively.

In the revised version, we have replaced the reference spectra in **Figure 2d** with the ones obtained from VO₂ and V₂O₃ powders. Besides, a sentence was added in the main text (**page 5, line 131**), as “By comparing with the EELS signals collected from commercially available VO₂ and V₂O₃ powders (**Supplementary Fig. S8**)”, and the information section (**page 10, line 280**) was also modified as “Commercial VO₂ and V₂O₃ powders (Sigma Aldrich 99.5% pure) were used for reference EELS spectra collection”. Moreover, **Figure R3** was added in the SI as **Figure S8**.

Figure R3. Comparison of EELS spectra between powders and c-facet thin films. **(a, b)** STEM-HAADF images (left panel) and EELS spectral images of (a) VO₂ and (b) V₂O₃ powders (right panel). **(c, d)** EELS spectra collected from powders (gray curves) and c-facet thin films before and after e-beam illumination (blue curves).

Q5. As claimed by the authors, the films used for e-beam illumination, AFM characterization, electrical and Raman measurements should be strained (18 nm case) but the ones used for TEM should be mostly relaxed (50 nm case). This structural change should have influence on the EELS if we are looking at very fine details across the thickness of the film, could the author provide depth dependent eels measurements across their pristine and irradiated films to verify the evolution of the V L_{2,3} and the O K edge across the thickness of the film? X-ray reciprocal space maps could shine more light on the exact strain state of the films for the thinner cases.

Response: We thank the reviewer for suggestions about the strain state of the thin film samples used in our measurements. We carried out EELS measurements across the thickness as the referee suggested. **Figure R4a** exhibits a STEM-HAADF image collected from the boundary of a c-facet VO₂ sample with both illuminated and pristine regions, where we can confirm the coexistence of both VO₂ (pristine region) and V₂O₃ (illuminated region) phases through FFT analysis. Then we extracted the depth-dependent EELS results at both regions (**Figures R4b, c**), which show no noticeable change of V L₃ peak (guided by black dotted line) within both pristine VO₂ and transformed V₂O₃, further demonstrating that the electronic states are homogeneous across the thin film. In the revised version, we have added **Figure R4** in the SI as **Figure S10**.

Finally, we want to point that the strain state of the SEM illuminated sample and STEM specimen is different due to the different sample geometry employed (**Figure R5**). Specifically, the thickness of SEM illuminated sample should be compared with the thickness of STEM specimen along the electron-beam. Importantly, because the observed phase

transformation can be triggered for both cases with the c-facet exposed to electron-beam, we believe the strain state does not play a major role for the observed facet-dependent phase transformation. Moreover, the facet dependent behavior indicates that the surface serves as a rate-limited factor during the phase transformation, in which thinner SEM sample would require shorter illuminated time to achieve the completed phase transformation, consistent with our experimental results as well. Furthermore, we chose the thicker thin films for the STEM measurements to minimize the influence of upper surface and lower interface to the observed phase transformation.

Figure R4. Depth-dependent EELS measurements of pristine and illuminated regions in c-facet VO_2 sample. (a) STEM-HAADF images collected at the boundary between illuminated and pristine regions. Corresponding FFT results are shown in the inset. The solid circles show the positions with EELS signal extracted. The scale bar is 5 nm. (b, c) Depth-dependent EELS measurements extracted from illuminated and pristine regions. The black dotted lines serve as a guideline.

Figure R5. Schematic illustrations of sample geometry for samples used for SEM and STEM experiments.

Q6. In figure 2d, could the author verify the absolute energy scale of their spectra, it seems to be shifted, on the other hand on this figure as well as figure S6 in the supplementary material it is very difficult to see subtle differences between the EELS spectra due to the large gap between the curves, for the purpose of fair comparison, it would be better to overlay the spectra and the authors should add the spectra of the pristine material in both orientations as there will be subtle differences that are important for the interpretation of the oxygen vacancies claim, as recently reported by Lee et al. (Science 2018)

Response: We thank the reviewer for the suggestion. As suggested, we overlaid the spectra as **Figure R6**, in which the changes of EELS spectra in c-facet sample can be clearly visualized. As pointed out by previous works (Lee D., et al *Science* **362** 1037-1040 (2018)), the extent of red shift of $V L_3$ peak and suppressed intensity of oxygen t_{2g} peak

can be correlated with the amount of oxygen vacancy (i.e., the valence state of V ion) in VO₂, therefore, the evident change in the c-facet sample implies the significant reduction of valence state in V ions through e-beam illumination. As for the a-facet sample, there is a much smaller change for V L-edge and O K-edge, indicating that only a small amount of the oxygen vacancy is created under same electron dose. We used reference spectra of V⁴⁺ and V³⁺ collected from commercial powder to fit the spectra of a-R, through which the concentration of oxygen vacancy is estimated as 3% (VO_{1.94}). In the revised version, **Figure R6** has been added in the SI to replace original **Figure S6**.

Also, in the revised **figure 2**, we added the spectra of pristine VO₂ thin films for both facets and overlaid the spectra before and after e-beam illumination, as shown in **Figure R7**. We also added a new sentence in the revised version (**page 5, line 136**). “On the other hand, the illuminated a-facet sample shows a much smaller change of peak position at L-edges and intensity of oxygen K-edges, which is consistent with the fact that the a-facet sample remains its pristine monoclinic phase with small amount (about 3%) of oxygen vacancy (**Fig. 1c, d**)”

Figure R6. Evolution of electronic structures under continuous e-beam illumination. The pronounced red shift of vanadium *L*-edges and dramatically suppressed *t*_{2g} peak of oxygen *K*-edges confirm the phase transformation into V₂O₃ for c-facet VO₂ sample, while the a-facet VO₂ sample maintains its pristine structure with the formation of only small amount (about 3%) oxygen vacancies.

Figure R7. Core-loss signals at vanadium L -edges and oxygen K -edges collected from c-facet and a-facet VO₂ samples before and after e-beam illumination, respectively. Reference spectra acquired from VO₂ and V₂O₃ powders are shown in the lower panel.

Q7. The authors said that knock-on damage can be excluded to be the source of the observation, but as the microscope used for the STEM measurement can be used from 60 to 300kV, it would be interesting to see if the authors can reproduce the same effect observed at 300kV (in ABF/HAAADF images and in EELS) at a lower voltage (60kV) where knock-on damage will be strongly reduced. In general the discussion of the e-beam induced phase transformation should be developed in more details and documented by a carefully researched references in the literature as electron beam damage is a very complex topic. For example, I am not convinced by the statement of the authors stating that they exclude radiolysis as it would amorphized the specimen.

Response: We greatly appreciate the valuable comments from the reviewer. Enlightened by this suggestion, we provided improved discussions in the revised version to further elaborate the mechanism of e-beam driven phase transformation.

- (i) Knock-on effect. We agree with the reviewer that the experiment carried out at 60 kV is very valuable to exclude the knock-on effect. However, we are unable to perform such measurement due to the limitation of experimental setup (the low-voltage mode has not been calibrated in the equipment we used as regular user). However, we want to note that similar effect was demonstrated here with a commercial SEM setup, in which the e-beam energy is 30 kV, and at this condition the knock-on effect would be suppressed due to both the much lower voltage and the absence of electron-exit surface [1].
- (ii) Beam heating. The origin of heating for TEM sample is due to the energy transfer from the beam to specimen via electron-phonon scattering. The total energy absorbed by the specimen from the electron beam and the

quality of the thermal contact between the specimen and support are two important factors determining the temperature change. The boundary condition for estimating the temperature gradient induced by electron illumination is complicated in real cases, especially with unknown quality of the thermal connection between specimen and support [2]. In this study, we could rule out the heating effect as main driving force based on the following two factors. First, we compared our results with *in-situ* heating XRD measurements shown in **Figure S13**, where no evident phase transformation to V_2O_3 is observed even at 170 °C. Second, similar phase transformation could be consistently observed even when we increased or decreased the illuminated area, indicating the thermal effect is not essential [3].

- (iii) Radiolysis. The radiolytic process in solid materials can result in the formation of stable defects within the bulk via nonradiative relaxation of excitation. Specifically, the decay of electronic excitations can break the chemical bonds, ultimately resulting in the defect formation [4-5]. The radiolysis has been used to describe defect formation and amorphization in halides and zeolites [6-7]. While in oxides, the radiolysis can also lead to the formation of Frenkel-type defect, which consists of O-O peroxy and O vacancy, and contributes to the amorphization of crystalline SiO_2 [8].
- (iv) Oxygen desorption. We appreciate the reviewer's comment on the radiolytic process, which inspires us take more thorough consideration about the effect of radiolysis. We realized that our initial understanding of radiolysis is incomplete, and we now further discuss about the effect of radiolysis on oxygen desorption. In transition metal oxides with multivalence of metal ions, the radiolytic process can also lead to oxygen desorption from the surfaces, caused by the transition from O^{2-} to O^0 or O^+ ions [9-10]. The subsequent reduction of metal ions and structural distortion occur primarily in high valent ionic materials, like TiO_2 , V_2O_5 and WO_3 , resulting in the formation of metallic monoxide phase at the surface [11-13].

Therefore, we modified the main mechanism of e-beam induced phase transformation from VO_2 to V_2O_3 , by combining the internal electric field formed by the positive charge background at the sample surface and the oxygen desorption effect, which creates a gradient chemical potential for oxygen ions between the bulk to the surface [14-15]. We note that these two mechanisms work cooperatively to facilitate the diffusion of oxygen vacancies into the bulk, ultimately driving the phase transformation.

In the revised manuscript, the corresponding sentences (**page 6, line 164**) were modified as “*The radiolysis and internal electrostatic field are hard to be decoupled, as the former is able to induce oxygen desorption from the surface³⁹, which creates a gradient chemical potential for oxygen ions between the bulk to the surface^{40, 41}; while the latter would naturally facilitate the migration of oxygen vacancy from surface toward the sample bulk due to positively charged surface⁴². Eventually, these two mechanisms contribute cooperatively to facilitate the observed phase transformation from VO_2 into the corresponding V_2O_3 (see **Supplementary Materials** for detailed discussion).*” Also, a new supplementary note “*1. Mechanism of e-beam induced phase transformation from VO_2 to V_2O_3* ” was added in the SI to detailly discuss the mechanism of e-beam induced phase transformation.

Reference

- [1] Egerton R. F., et al. *Ultramicroscopy* 110 991-997, 2010
- [2] Reimer L, *Transmission Electron Microscopy: Physics of Image Formation and Microanalysis*, 1989
- [3] Nan J, *Rep. Prog. Phys.* **79** 016501, 2016
- [4] Kabler M N, et al. *Phys. Rev. B* **60** 181948, 1978

- [5] Hobbs L W, et al. *J. Phys. Colloque* **41** 237, 1980
 [6] Sibley W A, et al. *Nucl. Instrum. Methods Phys. Res. B* **1** 419, 1984
 [7] Salamanca R L, et al. *Phys. Rev. B* **33** 2738–48, 1986
 [8] Hobbs L W, et al. *J. Non-Cryst. Solids* **182** 27–39, 1995
 [9] Ramsier R D, et al. *Surf. Sci. Rep.* **12** 246–376, 1991
 [10] Knotek M L, et al. *Phys. Rev. Lett.* **40** 964, 1978
 [11] Petford A K, et al. *Surf. Sci.* **172** 496–508, 1986
 [12] Fan H J, et al. *Ultramicroscopy* **31** 357–64, 1989
 [13] Smith D J, et al. *Ultramicroscopy* **23** 299–304, 1987
 [14] Park Y, et al. *Nano Lett.* **22** 9306–9312, 2022
 [15] Park Y P, et al. *Nat. Comm.* **11** 1401, 2020

Q8. Concerning the facet dependence of the phase transition, as the sample prepared for TEM is a cross section of the specimen, it would be adequate to mention the crystallographic orientation next to each STEM image. Could the authors provide an image of the cross section of the film after e-beam irradiation of the 18nm film taken under one of the irradiated area from c-facet features presented in figure 3?

Response: We thank the reviewer for the suggestion, and corresponding orientation of the STEM images were labelled in our revised manuscript and SI.

In this study, we also utilized FIB to prepare the cross-sectional samples of the c-facet film after e-beam illumination in SEM, as shown in **Figure R8a**. The element mapping confirms the thickness of our sample (**Figures R8b-d**). It is hard to directly examine the phase transformation based on structural features, as the atomic arrangements viewed along the orientation of V_2O_3 [0001] zone axis and VO_2 a-facet are very similar. To confirm whether the phase transformation is completed, we investigated the valence state of V ions using EELS. As exhibited in **Figure R8e**, the spectra of vanadium *L*-edge and oxygen *K*-edge of the thin film are consistent with the reference spectra of V_2O_3 , further supporting that the phase transformation is indeed driven by e-beam illumination.

Figure R8. Further confirmation of phase transformation from VO_2 to V_2O_3 induced by e-beam illumination. (a) TEM sample preparation for e-beam illuminated area. (b) STEM-HAADF image collected along [100] zone axis of TiO_2 . The green rectangle represents the area for EELS collection. (c) V and Ti EELS signal mapping. (d) EELS

signal profiles for V (blue curve) and Ti (yellow curve) captured along out-of-plane direction to confirm the thin film thickness. (e) V *L*-edges and O *K*-edges signals collected from the thin film. Spectra acquired from V₂O₃ powder is shown as a reference.

Q9. The sentence “Furthermore, we speculate that the e-beam induced positively charged background at the sample surface leads to an effective internal field as the thermodynamical driving force, boosting the diffusion of the negatively charged oxygen ions from the bulk toward sample surface.” Is quite confusing to me because this could be understood as oxygen vacancies starting to form in the bulk rather than at the surface which is not observed in the stem images.

Response: We apologize for the confusion caused by this sentence. Our original intention was to emphasize the positively charged background at the sample surface acts as the electrostatic driving force to facilitate the formation of oxygen vacancies at the sample surface, which then further diffuses into the bulk. To avoid further misunderstanding, we revised this sentence (**Page 7, line 197**) as “Furthermore, we speculate that the e-beam induced positively charged background at the sample surface leads to an effective internal electrostatic field to facilitate the formation of positively charged oxygen vacancies at the sample surface, which then migrates gradually into the bulk to trigger the phase transformation.”

Q10. When looking at Figure S1, only the top left panel shows Laue fringes around the film peak which attests to sharp interfaces and is a good proxy for high film quality. It is disappointing to see that the samples on TiO₂ (which have a better epitaxial match than those on Al₂O₃) don't show such finite thickness fringes, could the authors comment on this.

Response : We thank the reviewer for this careful inspection. We calibrated the XRD measurements and used fine step to measure the VO₂ thin films deposited on both a-facet and c-facet TiO₂ substrates. As shown in **Figure R9**, both them indeed show signature of Laue fringes. We note that this feature could be further improved by calibrating the growth conditions, especially the surface morphology of the substrates. However, we argue that this would not influence the main conclusion of this work, due to the following reasons. 1) Both the STEM and XRD measurements reveal that the films are single phase, and the lack of thickness fringes should be attribute to the interface quality. 2) Both the thin films grown on Al₂O₃ and TiO₂ show similar phase transformation through e-beam illumination, although they have different Laue fringes. In the revised version, **Figure R9** was added in the SI to replace original XRD result.

Figure R9. XRD measurements of two orientated VO₂ deposited on TiO₂ substrates.

Q11. Figure S5, also the TiO₂ substrate gets damaged? What is the reaction there? Some areas of the c-facet film are dark on the second image and white on the third, could the authors comment on this? The surface of the film seems strongly damaged on the second image but recovers in the third image, could the authors comment of this? I also see an increase of the amount of white spots between the first and second image of the a-facet but they become less visible on the third one, could the authors comment on this. This image has furthermore extremely low quality (similar to figures S4, S3, S6 and S7)

Response: To address the issue that whether the substrate gets damaged, we carried out more analysis of the TiO₂ substrate. **Figure R10a** displays STEM-HAADF images before and after e-beam illumination, which show no evident difference in the TiO₂ substrate after e-beam illumination. We also collected the EELS signals of Ti *L*-edges (**Figure R10b**) from these two regions and found that the Ti valence remains to be 4+ after e-beam illumination. Thus, we believe that the electron dose used in this work would not induce any significant change of the TiO₂. However, as the epitaxial relationship changes from VO₂/TiO₂ to V₂O₃/TiO₂, the interface is reasonable to have some structural or electronic reconstructions, which warrants further investigation.

As for the varying contrast of surface shown in some regions of the c-facet film, it could be caused by the image-filtering. And the varying amounts of white spots (surface absorption) in some regions of the a-facet film could be caused by the unnormalized brightness and contrast. We appreciate the reviewer's careful inspection. In the revised version, we have replaced the filtered image with the raw image and normalized their brightness and contrast (**Figure R11**).

Finally, the low quality of the STEM-HAADF images shown in SI is due to the compression during the conversion from .doc to .pdf file. We now updated these with new files.

Figure R10. Comparison of TiO₂ substrate before and after e-beam illumination. (a). STEM-HAADF image collected before (left panel) and after (right panel) illumination. (b) Ti *L*-edge signal collected from pristine (blue curve) and illuminated (red curve) regions.

Figure R11. Upper panel: Evolution of c-facet VO₂ samples under continuous e-beam illumination. Lower panel: Evolution of a-facet VO₂ under continuous e-beam illumination. The inset shows corresponding GPA results at different illuminating conditions. The thickness of illuminating area is determined by zero-loss of EELS, which is about 80 nm for both samples. The scale bar is 2nm.

Q12. It would be also interesting to know how the authors avoided contamination with such long electron beam irradiations as hydrocarbons contamination is a well known side effect of electron beam illumination.

Response: We thank the reviewer for this question. In order to avoid substantial hydrocarbon contamination on the sample surface during the measurements, we utilized plasma cleaning before the STEM experiment. We have now added a sentence in the method of revised manuscript (page 10, line 267) as “Plasma cleaning was employed for all TEM samples to remove the free carbon adsorbed on the surface”

Reviewer #3

In this manuscript, a method based on e-beam is exploited to induce chemical reduction in VO₂, which is accompanied by a structure transition into V₂O₃. I think that the evidence to show the transition into V₂O₃ is very indirect, which is not very convincing.

Q1. In Fig. 1d, the Raman spectrum for C-R is so broad and it looks almost like an amorphous material. Only the peak from the substrate remains sharp, and it is not really convincing to claim the existence of the crystal structure.

Response: We thank the reviewer for the critical comment. Regarding the Raman spectra, we now compared directly

the spectra of c-facet VO₂ after e-beam illumination with the result obtained from V₂O₃ powder, as illustrated in **Figure R12**. The good consistency observed between these two spectra confirms the transformed V₂O₃ phase in our experiment. In the revised manuscript, a new sentence was added in the main text (**page 6, line 148**) as “the characteristic Raman peaks in the illuminated c-facet sample could be attributed to the features of rhombohedral V₂O₃ (*Supplementary Fig. S11*)”, and **Figure R12** was added in the SI as **Figure S11**.

Figure R12. Comparison of Raman spectra taken at both illuminated c-facet sample and V₂O₃ powder.

Q2. It is also well-known that Vanadium oxide thin films could undergo various surface reconstruction. The EELS spectra of c-R exhibit a significant shift compared to the reference spectra of V₂O₃, which requires further investigations using other techniques like HAXPES.

Response: We appreciate the critical insight from the reviewer about the surface reconstruction of vanadium oxide thin films. In our study, particularly the TEM measurements, the thickness of sample is about 80 nm determined by zero-loss peak (**Figure R13**), in which the surface reconstruction unlikely affects the observed phase transformation. To make a reliable comparison, we now collected reference EELS spectra from commercial V₂O₃ powder to replace the original spectra adapted from reference. As shown in **Figure R14**, now the c-R spectra are nicely consistent with the referred V₂O₃ spectra. In the revised manuscript, the VO₂ and V₂O₃ reference spectra in **Figure 2d** was replaced by the data collected from commercial powders.

Moreover, the transformation from VO₂ to V₂O₃ in c-facet sample after e-beam illumination can also be confirmed by the atomic-resolved STEM image (**Figures R15a, b**). We performed image simulation using the rhombohedral V₂O₃ structure and the obtained simulated image matches nicely with our experimental image (**Figures R15c, d**), strongly confirming the transformation into V₂O₃. We used **Figure R15** to replace original **Figure 3f** in the revised manuscript and added a new sentence (**Page 5, line 141**) as “Combined with the high angle annular dark field (STEM-HAADF) and annular bright field (STEM-ABF) images, the atomic positions of vanadium and oxygen ions are determined, which together with the simulation results (**Fig. 2f**), provide compelling evidence that the illuminated area transforms from monoclinic VO₂ to rhombohedral V₂O₃.”

Figure R13. Estimate the sample thickness based on the zero-loss peak.

Figure R14. Comparison of EELS spectra taken from V_2O_3 powder and c-facet VO_2 after e-beam illumination.

Figure R15. Comparison between experimental and simulated images of V_2O_3 . (a, b) STEM-HAADF and STEM-ABF images collected from illuminated c-facet VO_2 . The dashed line represents the interface between TiO_2 substrate and V_2O_3 . (c, d) Left panel: The zoom-in HAADF and ABF images of the thin film. Right panel: Simulated HAADF and ABF image using the rhombohedral V_2O_3 as the input.

Q3. For the DFT results, the criterion is set to be 0.02 eV/Å, which seems to be higher than typical recommended value of 0.01. Moreover, details about how the structural optimization are missing. For example, are all atoms allowed to move? Is volume allowed to change? What optimization schemes in VASP are used?

Response: We thank the reviewer for this valuable comment. Following the suggestion, we repeated the calculations with the change of force convergence from 0.02 eV/Å to 0.01 eV/Å, and the obtained results show no significant difference. In the revised manuscript, we have updated the results with the data obtained with the new criterion of 0.01 eV/Å.

During the structural optimization, both the cell volume and atom positions were allowed to move during the self-consistent calculation. While during the calculations of the surface formation energy, NEB, and interface formation energy calculation, the cell volume was fixed, while all atoms were allowed to relax. In the calculations, the electronic minimization adopted the blocked Davidson iteration scheme, and the ionic relaxation was used in the conjugate gradient algorithm with IBRION = 2.

In the revised manuscript, we have modified the method section (**page 10, line 284**) as following, *“In the calculations, we used the Perdew-Burke-Ernzerhof (PBE) formulation of the generalized-gradient approximation (GGA) for the exchange-correlation function using projector augmented-wave method (PAW)the climbing image nudged elastic band (CI-NEB) method for migration barrier calculation and interface formation energy calculations. During the structural optimization, both the cell volume and atom positions are allowed to move during the self-consistent calculation. While during the calculations of the surface formation energy, NEB, and interface formation energy calculation, the cell volume is fixed, while all atoms are allowed to relax. In the calculations, the electronic minimization adopted the blocked Davidson iteration scheme, and the ionic relaxation was used in the conjugate gradient algorithm.”*

Q4. V₂O₃ is known to have magnetic phase transition. It will be necessary to have some magnetic property analysis if the authors want to claim the existence of V₂O₃.

Response: We thank the reviewer for this constructive suggestion and pointing to a promising new research topic. As the current results (STEM analysis, Raman spectra, EELS spectra, transport analysis) all confirmed the phase transformation into V₂O₃, we would leave the study of antiferromagnetic state for a future study. Indeed, the construction of antiferromagnetic pattern (V₂O₃) embedded in the paramagnetic matrix (VO₂) through this newly developed approach sounds like an exciting proposal, and could potentially lead to some intriguing discoveries in the field of spintronics, and we will devote further efforts along this.

In summary, the present experimental data are not convincing enough to support the claim of the paper.

Response: Based on the critical comments from the referee, we have further confirmed the phase transformation into V₂O₃ through e-beam illumination. We hope the referee is now satisfied with the revision.

REVIEWERS' COMMENTS

Reviewer #1 (Remarks to the Author):

Dear Editor, in my view, the authors have answered and addressed all questions and comments raised by the referees adequately and thoroughly. The only thing I noticed that the authors in answering Q10 (Ref2) wrongly attributed the limited intensity of the Laue fringes to 'and the lack of thickness fringes should be attribute to the interface quality.' The fringes present themselves more pronounced if the crystal coherence in the films is better so apparently in the films on TiO₂ coherence is lower. This should not affect their overall outcome. In the question on magnetic properties, it would be nice to add but I agree that it is also not necessary for the main conclusion of the paper. Therefor I deem the manuscript ready for publication.

Reviewer #2 (Remarks to the Author):

The authors answered all my queries and I therefore agree with the manuscript in its revised state

Reviewer #3 (Remarks to the Author):

I have reviewed the revised manuscript and new data. I am satisfied with the quality of the current data presented in the manuscript, thus I don't have reservation anymore.

While I am convinced the existence of V₂O₃, lacking of the indication of magnetic properties prevents me from making a strong recommendation for publication on Nature Communications. To make it clear, my position is that while I don't object to see this manuscript published on Nature communications, I don't feel strongly that this manuscript should be published either.

Reviewer #1

Dear Editor, in my view, the authors have answered and addressed all questions and comments raised by the referees adequately and thoroughly. The only thing I noticed that the authors in answering Q10 (Ref2) wrongly attributed the limited intensity of the Laue fringes to 'and the lack of thickness fringes should be attribute to the interface quality.' The fringes present themselves more pronounced if the crystal coherence in the films is better so apparently in the films on TiO₂ coherence is lower. This should not affect their overall outcome. In the question on magnetic properties, it would be nice to add but I agree that it is also not necessary for the main conclusion of the paper. Therefor I deem the manuscript ready for publication.

Repones: We would like to express our appreciation for the time and effort the referees have dedicated to reviewing our reply. We sincerely thank the referees for their insightful comments on the Laue fringes and consideration for publication.

Reviewer #2

The authors answered all my queries and I therefore agree with the manuscript in its revised state

Repones: We sincerely thank the referees for their time in reviewing our reply and recommending our paper for publication.

Reviewer #3

I have reviewed the revised manuscript and new data. I am satisfied with the quality of the current data presented in the manuscript, thus I don't have reservation anymore.

While I am convinced the existence of V₂O₃, lacking of the indication of magnetic properties prevents me from making a strong recommendation for publication on Nature Communications. To make it clear, my position is that while I don't object to see this manuscript published on Nature communications, I don't feel strongly that this manuscript should be published either.

Repones: We sincerely thank the referees for recognizing the high-quality of current data and recommending our work for publication.